# Sen1 has unique structural features grafted on the architecture of the Upf1-like helicase family

Bronislava Leonaitė[1,2,†], Zhong Han[3,4,†], Jérôme Basquin[1], Fabien Bonneau[1], Domenico Libri[3] ,
Odil Porrua[3,*] & Elena Conti[1,**]

## Abstract

The superfamily 1B (SF1B) helicase Sen1 is an essential protein that plays a key role in the termination of non-coding transcription in yeast. Here, we identified the ~90 kDa helicase core of *Saccharomyces cerevisiae* Sen1 as sufficient for transcription termination *in vitro* and determined the corresponding structure at 1.8 Å resolution. In addition to the catalytic and auxiliary subdomains characteristic of the SF1B family, Sen1 has a distinct and evolutionarily conserved structural feature that "braces" the helicase core. Comparative structural analyses indicate that the "brace" is essential in shaping a favorable conformation for RNA binding and unwinding. We also show that subdomain 1C (the "prong") is an essential element for 5′-3′ unwinding and for Sen1-mediated transcription termination *in vitro*. Finally, yeast Sen1 mutant proteins mimicking the disease forms of the human orthologue, senataxin, show lower capacity of RNA unwinding and impairment of transcription termination *in vitro*. The combined biochemical and structural data thus provide a molecular model for the specificity of Sen1 in transcription termination and more generally for the unwinding mechanism of 5′-3′ helicases.

**Keywords**  non-coding transcription; RNA helicases; transcription termination
**Subject Categories**  RNA Biology; Structural Biology
**The EMBO Journal (2017) 36: 1590–1604**

## Introduction

In yeast, there are two major transcription termination pathways. In the case of canonical protein-coding genes, Pol II normally terminates transcription via the cleavage and polyadenylation factor (CPF) complex, yielding stable mature mRNAs that are then exported to the cytoplasm (reviewed in Mischo & Proudfoot, 2013). In the case of non-coding RNAs, such as cryptic unstable transcripts (CUTs) and small nucleolar RNAs (snoRNAs), Pol II terminates transcription via a non-canonical pathway that is coupled to RNA degradation (reviewed in Jensen *et al*, 2013). This non-canonical termination pathway depends on the Nrd1-Nab3-Sen1 (NNS) complex (reviewed in Arndt & Reines, 2015; Porrua & Libri, 2015a). Nrd1 and Nab3 form a heterodimer (Carroll *et al*, 2007) that underpins the substrate specificity of the NNS complex (Wlotzka *et al*, 2011; Porrua *et al*, 2012; Schulz *et al*, 2013). Nrd1-Nab3 also recruits the Trf4 subunit of TRAMP (Tudek *et al*, 2014), a major cofactor of the RNA-degrading exosome in the nucleus (LaCava *et al*, 2005; Vanacova *et al*, 2005; Wyers *et al*, 2005). TRAMP polyadenylates the 3′ end of its RNA substrates and feeds them to the nuclear exosome, resulting in the complete or partial 3′-5′ degradation of CUTs and snoRNAs, respectively (Allmang *et al*, 1999; Wyers *et al*, 2005). While Nrd1-Nab3 couples the NNS complex to RNA degradation, Sen1 is the key enzyme in the transcription termination reaction (Porrua & Libri, 2013). Sen1 has also been shown to be involved in termination of short protein-coding genes (Steinmetz *et al*, 2006), and inactivation of Sen1 leads to the accumulation of R-loops (RNA:DNA hybrids that form during transcription when the nascent RNA invades the DNA template) (Mischo *et al*, 2011).

Sen1 is an RNA/DNA helicase and is the only evolutionarily conserved subunit of the NNS complex. The human orthologue of yeast Sen1, senataxin (SETX), is associated with neurological pathologies: recessive mutations in the *SETX* gene cause ataxia with oculomotor apraxia type 2 (AOA2) and dominant mutations provoke amyotrophic lateral sclerosis type 4 (ALS4) (reviewed in Bennett & La Spada, 2015). Disease mutations cluster in the two most conserved regions of SETX, the N-terminal domain and the helicase domain. Like its yeast orthologue, SETX has been assigned functions in transcription termination and in the control of R-loop formation (Suraweera *et al*, 2009; Skourti-Stathaki *et al*, 2011; Zhao *et al*, 2016).

Sen1 belongs to the superfamily 1B (SF1B) Upf1-like family of helicases together with Upf1 and IGHMBP2. Structural studies of Upf1 and IGHMBP2 have shown the presence of a common domain organization (Cheng *et al*, 2007; Clerici *et al*, 2009; Chakrabarti

1   Max Planck Institute of Biochemistry, Munich, Germany
2   Graduate School of Quantitative Biosciences, Ludwig-Maximilians-University, Munich, Germany
3   Institut Jacques Monod, Centre Nationale pour la Recherche Scientifique (CNRS), UMR 7592, Université Paris Diderot, Paris, France
4   Université Paris-Saclay, Gif sur Yvette, France
   *Corresponding author. Tel: +33 1 57 27 80 35; E-mail: odil.porrua@ijm.fr
   **Corresponding author. Tel: +49 89 85 78 36 02; E-mail: conti@biochem.mpg.de
   †These authors contributed equally to the work

   

*et al*, 2011; Lim *et al*, 2012). SF1B RNA helicases contain two RecA domains (RecA1 and RecA2) with the classical helicase motifs involved in nucleic acid binding and ATP hydrolysis. Helicases of this family also contain two SF1B-specific subdomains (1B and 1C) that modulate RNA binding (Cheng *et al*, 2007; Clerici *et al*, 2009; Chakrabarti *et al*, 2011; Lim *et al*, 2012). SF1B helicases bind nucleic acids with the same polarity as all other RNA-dependent ATPases, that is, with the 3′ end at RecA1 and the 5′ end at RecA2 (reviewed in Pyle, 2008; Ozgur *et al*, 2015). However, the directionality of duplex unwinding of the SF1B superfamily is opposite to that of processive SF2 helicases, which unwind duplexes in the 3′-5′ direction (Büttner *et al*, 2007). For example, Upf1 has been shown to be a highly processive 5′-3′ RNA helicase (Bhattacharya *et al*, 2000; Fiorini *et al*, 2015). Similarly, Sen1 uses ATP hydrolysis to unwind RNA or DNA duplexes in the 5′-3′ direction (Kim *et al*, 1999; Martin-Tumasz & Brow, 2015).

Sen1 is expected to have a similar domain organization as compared to Upf1 and IGHMBP2 and a similar 5′-3′ unwinding mechanism. However, Sen1 also has a distinct function, namely the ATPase-dependent ability of promoting transcription termination *in vitro* (Porrua & Libri, 2013). In this work, we used biochemical and structural approaches to dissect the elements that underpin the general 5′-3′ unwinding and the distinctive properties of Sen1. We demonstrate the existence of features that are specific to Sen1 and integrate structural knowledge into a refined model for 5′-3′ unwinding and transcription termination.

## Results and Discussion

### Identification of the active helicase core of *Saccharomyces cerevisiae* Sen1

*Saccharomyces cerevisiae* Sen1 is a multi-domain protein of about 250 kDa (2231 residues). Analysis of the Sen1 amino acid sequence by secondary structure and fold recognition programs HHpred (Söding *et al*, 2005) and Phyre2 (Kelley *et al*, 2015) predicted the presence of an α-helical region (amino acids 1–975) at the N-terminus followed by a Upf1-like helicase region and a low-complexity segment of roughly 300 residues at the C-terminus (Fig 1A). To obtain a soluble fragment of Sen1 encompassing the helicase core, we subcloned a fragment of Sen1 cDNA coding for residues 976–1,904 into a vector for bacterial expression. After purification, we noticed that the Sen1 fragment was smaller than expected (Fig EV1A). Mass-spectrometry analyses suggested that endogenous protease digestion had likely occurred at the Sen1 N-terminus during purification.

For structural studies, we expressed and purified a fragment of Sen1 encompassing residues 1,095–1,904 (hereafter referred to as Sen1$_{Hel}$), with the shorter N-terminus previously identified by the Brow's group (Martin-Tumasz & Brow, 2015). Sen1$_{Hel}$ showed levels of RNA-dependent ATPase activity similar to those of full-length Sen1 purified from yeast (ySen1 FL). A mutant version with the E1591Q substitution in the highly conserved helicase motif II, which is essential for ATP hydrolysis, was inactive (Fig 1A). Next, we analyzed the helicase activity of Sen1$_{Hel}$ by performing duplex unwinding assays. As a substrate, we used a 19-nt RNA:DNA duplex harboring a 25-nt single-stranded RNA extension at either the 5′-end or the 3′-end. Similar to the full-length protein, Sen1$_{Hel}$ displayed significant duplex unwinding activity on the substrate containing a 5′ single-strand overhang (Fig 1B), while no activity was detected on the substrate containing a single-stranded extension at the 3′ end (Fig EV1B), consistent with 5′-3′ helicase activity. We noted that Sen1$_{Hel}$ has similar unwinding properties as compared to the Sen1 fragment that had previously been characterized by the Brow's group (Martin-Tumasz & Brow, 2015).

Next, we tested whether recombinant Sen1$_{Hel}$ retains the capability of the full-length protein to terminate transcription *in vitro* (Porrua & Libri, 2013). We assembled ternary elongation complexes (ECs) in a promoter-independent manner using purified RNA Pol II, DNA transcription templates, and a short RNA oligonucleotide primer that forms a 9-bp duplex with the template strand and occupies the active center of the polymerase (Fig 1C, left panel). We biotinylated the non-template strand to allow the association of ECs with streptavidin beads and the subsequent separation of Pol II molecules (and associated transcripts) engaged in transcription from

**Figure 1. A recombinant version of the *Saccharomyces cerevisiae* Sen1 helicase core retains the main biochemical properties of the full-length protein.** ▶

A   Analysis of RNA-dependent ATPase activity of Sen1 proteins. Top: Schematic diagram of full-length Sen1 purified from yeast (ySen1 FL), and recombinant Sen1$_{Hel}$ and Sen1$_{Hel}$ E1591Q mutant. An asterisk denotes the presence of a mutation in the helicase domain. Bottom, left: SDS–PAGE analysis of the purified proteins used in these assays (M: molecular weight marker). 2 pmol of ySen1 FL and 25 pmol of Sen1$_{Hel}$ proteins were loaded. Bottom, right: Graphical representation of the ATP hydrolyzed by the different Sen1 proteins as a function of time. Values represent the average and standard deviation (SD) from three independent experiments.

B   Time course analysis of the ATP-dependent 5′-3′ duplex unwinding activity of Sen1 proteins. Reactions contained 5 nM of Sen1 and 2 nM of substrate. An RNA:DNA duplex composed of a 44-mer RNA annealed to a 19-mer DNA molecule to provide a 5′-end 25-nt single-strand overhang was used as the substrate (see Appendix Table S1 for sequence details). The asterisk (*) denotes the presence of a FAM at the 5′ end of the DNA. The first lanes correspond to heat-denatured (95°C) samples, and the last lanes are control reactions incubated with Sen1 proteins in the absence of ATP. The graph on the right shows the fraction of duplex unwound as a function of time. Data were fitted with Kaleidagraph to the Michaelis–Menten equation. Values represent the average and standard deviation (SD) from three independent experiments.

C   *In vitro* transcription termination (IVTT) assays with 20 nM of ySen1 FL and 40 nM of Sen1$_{Hel}$ proteins. Left: Scheme of an IVTT assay. Ternary ECs composed of Pol II, fluorescently labeled nascent RNA, and DNA templates are assembled and attached to streptavidin beads via the 5′ biotin of the non-template strand to allow subsequent separation of beads-associated (B) and supernatant (S) fractions. An asterisk (*) denotes the presence of a FAM at the 5′ end of the RNA. The transcription template contains a G-less cassette followed by a G-stretch in the non-template strand. After adding ATP, UTP, CTP mix, Pol II transcribes until it encounters the G-rich sequence. Sen1 dissociates ECs paused at the G-stretch and releases Pol II and the associated transcripts to the supernatant. Right: PAGE analysis of RNAs from a representative IVTT assay. The fraction of transcripts released from ECs stalled at the G-stretch is used as a measure of the termination efficiency. Representative gel of one out of two independent experiments (values of RNA released in both experiments can be found in the corresponding source data file).

Source data are available online for this figure.

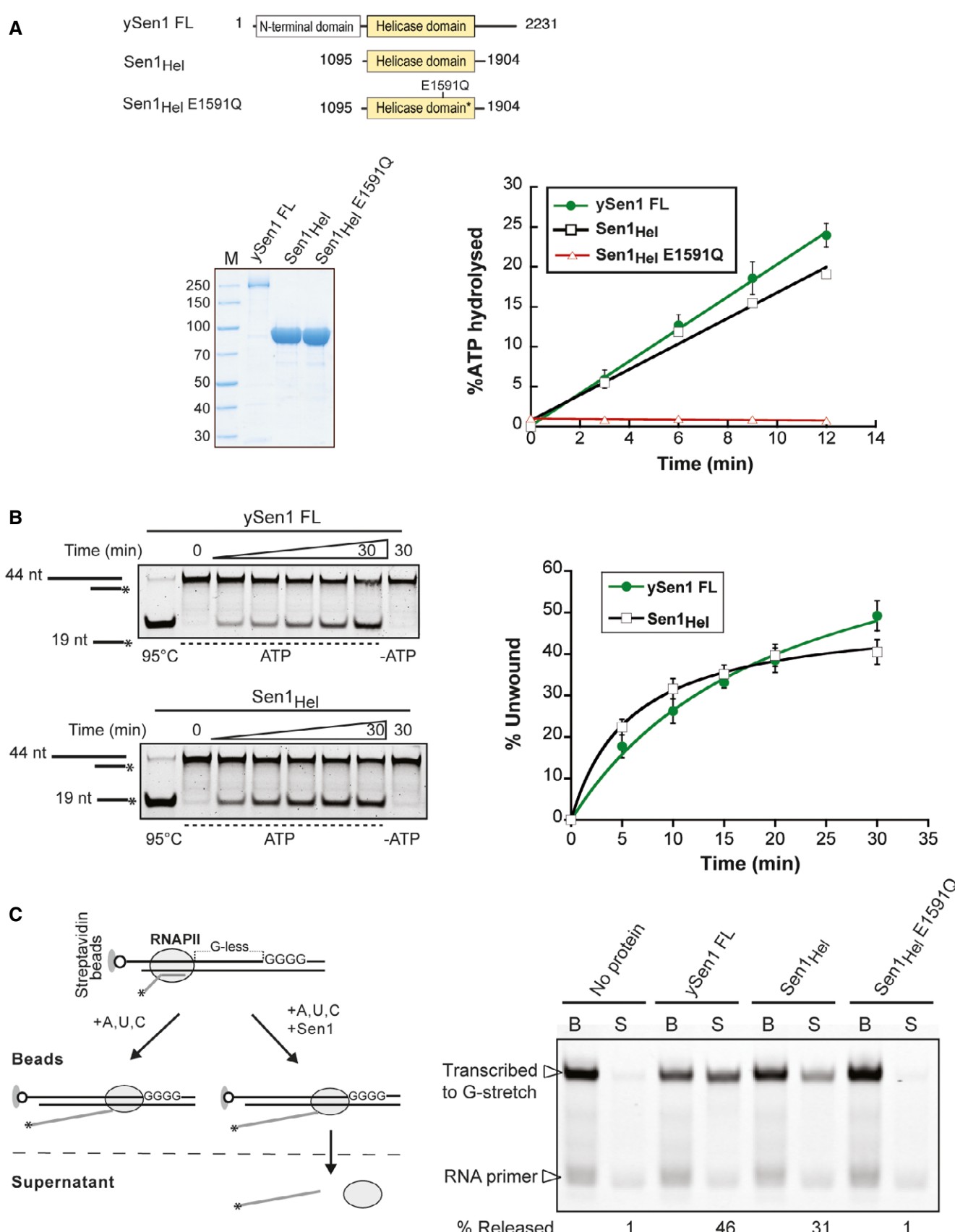

**Figure 1.**

those that have been released from the DNA templates after transcription termination. In order to assess the capacity of Sen1 proteins to elicit termination, we monitored the efficiency of release of nascent RNA into the supernatant. Similar to the full-length protein, Sen1$_{Hel}$ promoted the release of a significant fraction of nascent transcripts (Fig 1C, right panel). This activity was dependent on the integrity of the Sen1$_{Hel}$ active site, as no release was observed with the Sen1$_{Hel}$ E1591Q mutant (Fig 1C, right panel). We concluded that the ~90 kDa helicase core is essentially responsible for the transcription termination properties of Sen1 *in vitro*.

## Overall structure of the helicase core of yeast Sen1

We crystallized Sen1$_{Hel}$ in the presence of ADP and determined the structure by single-wavelength anomalous dispersion (SAD) phasing using the signal from the sulfur atoms in the native protein. The structure was refined at 1.8 Å resolution with $R_{free}$ of 18%, $R_{factor}$ of 15%, and good stereochemistry (Table 1) (Fig EV1C). Overall, Sen1$_{Hel}$ has a domain organization similar to that of the helicase core of Upf1 (Upf1$_{Hel}$, also known as Upf1-ΔCH) (Cheng *et al*, 2007; Clerici *et al*, 2009; Chakrabarti *et al*, 2011) as well as IGHMBP2 (IGHMBP2$_{Hel}$) (Lim *et al*, 2012) (Fig 2). In the Sen1$_{Hel}$-ADP structure, the two RecA domains are positioned side by side, separated by a cleft about 10 Å wide (Fig 2). In this open conformation, RecA2 is rotated about 30° from the position it acquires in the closed conformation that is typical of helicases in the active RNA-ATP-bound state (Linder & Lasko, 2006; Pyle, 2008). Two mutations shown to affect the function of Sen1 *in vivo* map to the RecA domains and are likely to cause partial unfolding of the protein due to unfavorable electrostatic clashes (G1747D, DeMarini *et al*, 1992 and E1597K, Steinmetz & Brow, 1996).

The ADP nucleotide binds at the bottom of the cleft and interacts directly with the RecA1 domain (Figs 2 and EV2A). Similarly to what has previously been observed in the structures of nucleotide-bound Upf1$_{Hel}$ (Chakrabarti *et al*, 2011), the adenine ring is sandwiched between an apolar surface of RecA1 and an aromatic residue (Tyr1655) that is present in the linker connecting RecA1 to RecA2 and is part of motif IIIa (Fairman-Williams & Jankowsky, 2012). In addition, the conserved side chain of Gln1339 forms a bidentate hydrogen-bond interaction with the N6 and N7 moieties of the adenine ring (Fig EV2A). A similar Gln-based specificity determinant for adenine nucleotides was originally identified in the so-called Q motif of DEAD-box proteins (Cordin *et al*, 2004). Although at the sequence level the corresponding Q motif of Upf1-like helicases is also present upstream of motif I, at the three-dimensional level it forms part of a different structural element as compared to the Q motif of DEAD-box proteins and is instead similar to the Q motif found in the Ski2-like family of DExH-box proteins (Sengoku *et al*, 2006; Jackson *et al*, 2010; Weir *et al*, 2010).

Sen1$_{Hel}$ also contains two accessory subdomains that extend on the surface of RecA1: a ~160-residue insertion known as subdomain 1B and a 120-residue insertion known as subdomain 1C. Subdomain 1B contains two antiparallel helices that pack against each other and against the side of RecA1 with extensive hydrophobic interactions, forming the so-called "stalk" (Chakrabarti *et al*, 2011) (Fig 2). At the top of the "stalk", subdomain 1B

features a β-barrel fold (the "barrel"), which hovers over RecA1. Subdomain 1C is also formed by α-helices, forming a prong-like feature. As outlined below, the "stalk", the "barrel", and the "prong" show specific differences when comparing Sen1$_{Hel}$ to Upf1$_{Hel}$ and IGHMBP2$_{Hel}$. The major difference from other known SF1B helicases, however, is the presence in Sen1$_{Hel}$ of a ~50-residue N-terminal segment that we refer to as the "brace" (Fig 2, left panel).

**Table 1. Crystallography statistics.**

| Data set | Sen1$_{Hel}$ native | Sen1$_{Hel}$ S-SAD |
|---|---|---|
| Data collection | | |
| Space group | P 21 21 2 | P 21 21 2 |
| Unit cell (a, b, c in Å) | 90.285, 171.944, 69.094 | 90.2, 171.66, 68.85 |
| Wavelength (Å) | 1.00 | 2.095 |
| Resolution range (Å) | 48.39–1.787 (1.851–1.787) | 85.83–2.145 (2.221–2.144) |
| Total reflections | 680,302 (29,401) | 1,643,000 |
| Unique reflections | 100,766 (2,170) | 114,276 (8,032) |
| Multiplicity | 13.2 (13.5) | |
| Completeness (%) | 98.27 (95.07) | 93.9 (68.81) |
| Mean I/sigma(I) | 18.8 (1.6) | 22.47 (1.2) |
| Wilson B-factor | 29.6 | 30.57 |
| R-merge | 0.085 (1.500) | N/D |
| R-meas | 0.092 | 0.086 |
| CC1/2 | 0.999 (0.610) | 0.999 (0.69) |
| CC* | 1 (1) | 1 (1) |
| Refinement | | |
| R-work (%) | 15.28 | |
| R-free (%) | 18.36 | |
| Number of non-hydrogen atoms | 6,970 | |
| Macromolecules | 5,543 | |
| Ligands | 337 | |
| Water | 1,090 | |
| Protein residues | 682 | |
| RMS (bonds) | 0.011 | |
| RMS (angles) | 1.39 | |
| Ramachandran favored (%) | 98 | |
| Ramachandran outliers (%) | 0 | |
| Clashscore | 10.86 | |
| Average B-factor | 49.2 | |
| Macromolecules | 42 | |
| Ligands | 100.50 | |
| Solvent | 66.4 | |

Statistics for the highest-resolution shell are shown in parentheses.

## Sen1 is a SF1B helicase with distinct structural features

Structural comparisons of Sen1$_{Hel}$ with Upf1$_{Hel}$ and IGHMBP2$_{Hel}$ reveal several distinct features in the accessory subdomains of Sen1$_{Hel}$ (Fig 2). First, the ordered portion of the "prong" is shorter with respect to Upf1$_{Hel}$ and IGHMBP2$_{Hel}$. Second, the "barrel" has a more elaborate topology with respect to Upf1$_{Hel}$ and IGHMBP2$_{Hel}$, with additional helical turns. Perhaps more importantly, the "barrel" is connected to the "stalk" helices by short linkers as compared to Upf1$_{Hel}$ and IGHMBP2$_{Hel}$. The short linkers appear to restrict the conformational space that the Sen1$_{Hel}$ "barrel" domain can sample. This spatial restriction is further compounded by the interactions with the N-terminal "brace" described below.

The "brace" (residues 1,097–1,149) fastens three different structural features of the helicase core, namely RecA1, the "stalk", and the "barrel". A first short α-helix (α1) inserts aliphatic side chains (Leu1109 and Arg1108) into a hydrophobic surface groove formed between the RecA1 domain (Ala1573, Ala1578, and Tyr1606) and a "stalk" helix (Tyr1303) (Fig 3A). The polypeptide chain then continues with a second α-helix (α2) sandwiched between the "stalk" helices and the "barrel". Hydrophobic residues on one side of helix α2 (Leu1116, Ile1120, and Trp1123) make apolar interactions with residues of the "stalk" (with Ile1291 and with Leu1162, Trp1166, and Leu1169, respectively) (Fig 3B). Hydrophobic residues on the other side of helix α2 (Tyr1117 and Leu1121) are engaged in van der Waals interactions with aliphatic side chains of the "barrel" (with Leu1216, Leu1244, and Lys1246). After another hydrophobic interaction with the "barrel" (Tyr1125 with Val1218 and Val1283), the "brace" makes an 180° turn via the clustering of Pro1132 with

Trp1123 and Trp1166. It then continues to connect to the ascending helix of the "stalk" via van der Waals interactions (Val1143 with Phe1147 with Tyr1153) (Fig 3B). Overall, the "brace" buries 2,400 Å$^2$ of the Sen1$_{Hel}$ surface area with evolutionarily conserved interactions (Fig 3C).

The "brace" appears to stabilize the overall fold of the protein. *In vitro*, deletions of the N-terminal 1,128–1,149 residues resulted in an insoluble protein, likely because removal of the "brace" led the hydrophobic residues on RecA1, the "stalk" and the "barrel" to be exposed to solvent (B Leonaite, E Conti, unpublished observations). Consistently, a deleted variant of Sen1 lacking the N-terminal 1,134 residues does not support yeast viability, while deletion of the N-terminal 1,088 residues (which leaves the "brace" intact) results in termination defects but does not lead to lethality (Chen *et al*, 2014). Moreover, a W1166S mutant has been shown to be defective *in vivo* (Chen *et al*, 2014). Thus, the "brace" is an important element for Sen1 function both *in vitro* and *in vivo*.

The structural analysis of Sen1$_{Hel}$ suggests that the "brace" firmly connects the "barrel" to the "stalk" helices. Although the conformation we observe might also be partly stabilized by lattice contacts, the extensive intramolecular interactions mediated by the "brace" and the short connections described above appear to genuinely restrain the position of the "barrel" on top of the RecA1 domain. We compared the position of the "barrel" of Sen1$_{Hel}$ with that of other SF1B helicases (Figs 4A and EV2B). In the apo structure of UPF1$_{Hel}$, the "barrel" is in close contact with RecA1 and interferes with the RNA-binding surface (Cheng *et al*, 2007). Upon RNA binding, the Upf1 "barrel" moves away from RecA1, effectively sandwiching part of the nucleic acid (Chakrabarti *et al*, 2011). In the apo structure of

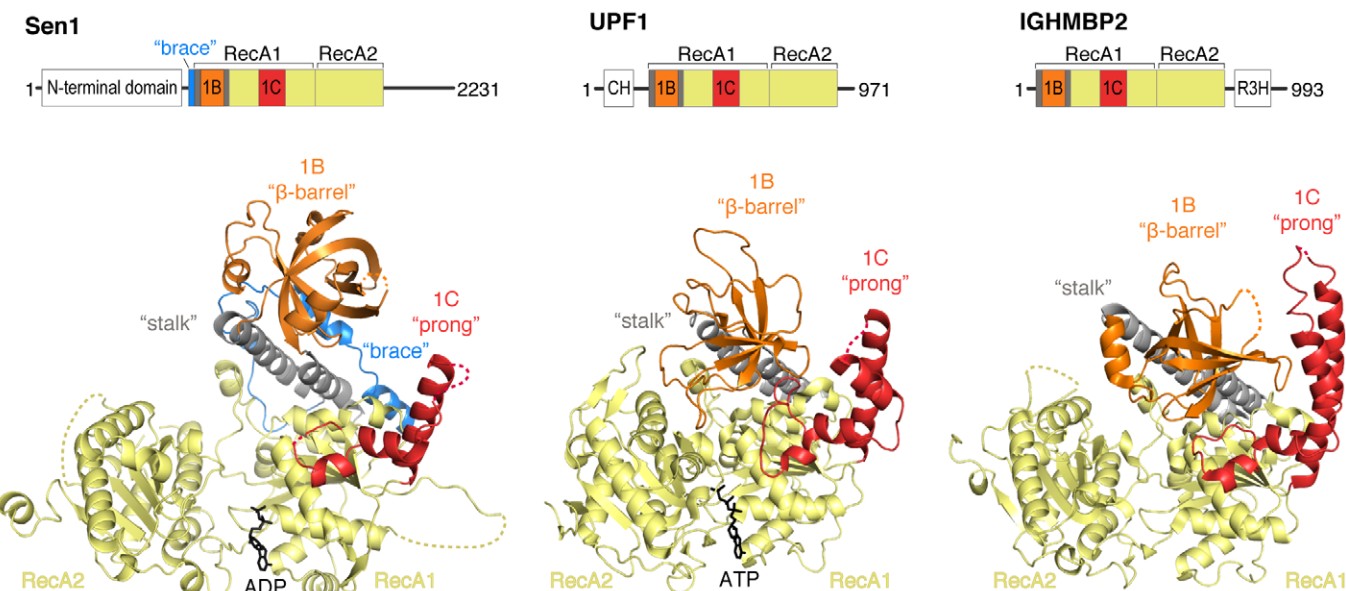

**Figure 2. Common and unique structural features of the Sen1 helicase core.**
Structures of yeast Sen1$_{Hel}$-ADP (left), Upf1$_{Hel}$-AMPPNP (middle, PDB: 2GJK, Cheng *et al*, 2007), and IGHMBP2$_{Hel}$ (right, PDB: 4B3F, Lim *et al*, 2012) determined in the absence of RNA are shown in a similar orientation after optimal superposition of their respective RecA1 domains (on the right in this front-view orientation). Dotted lines indicate disordered loops not modeled in the present structure. On top, there is a scheme with the domain organization of the full-length proteins, with predicted structured and unstructured regions shown as rectangles and lines, respectively. The fragments crystallized are highlighted in color. The RecA1 and RecA2 domains are in yellow, the "stalk" in gray, subdomain 1B (the "barrel") in orange and subdomain 1C (the "prong") in red. In the case of Sen1$_{Hel}$, the N-terminal "brace" is shown in blue. CH domain, cysteine and histidine rich domain; R3H domain, Arg-x-x-His motif containing domain.

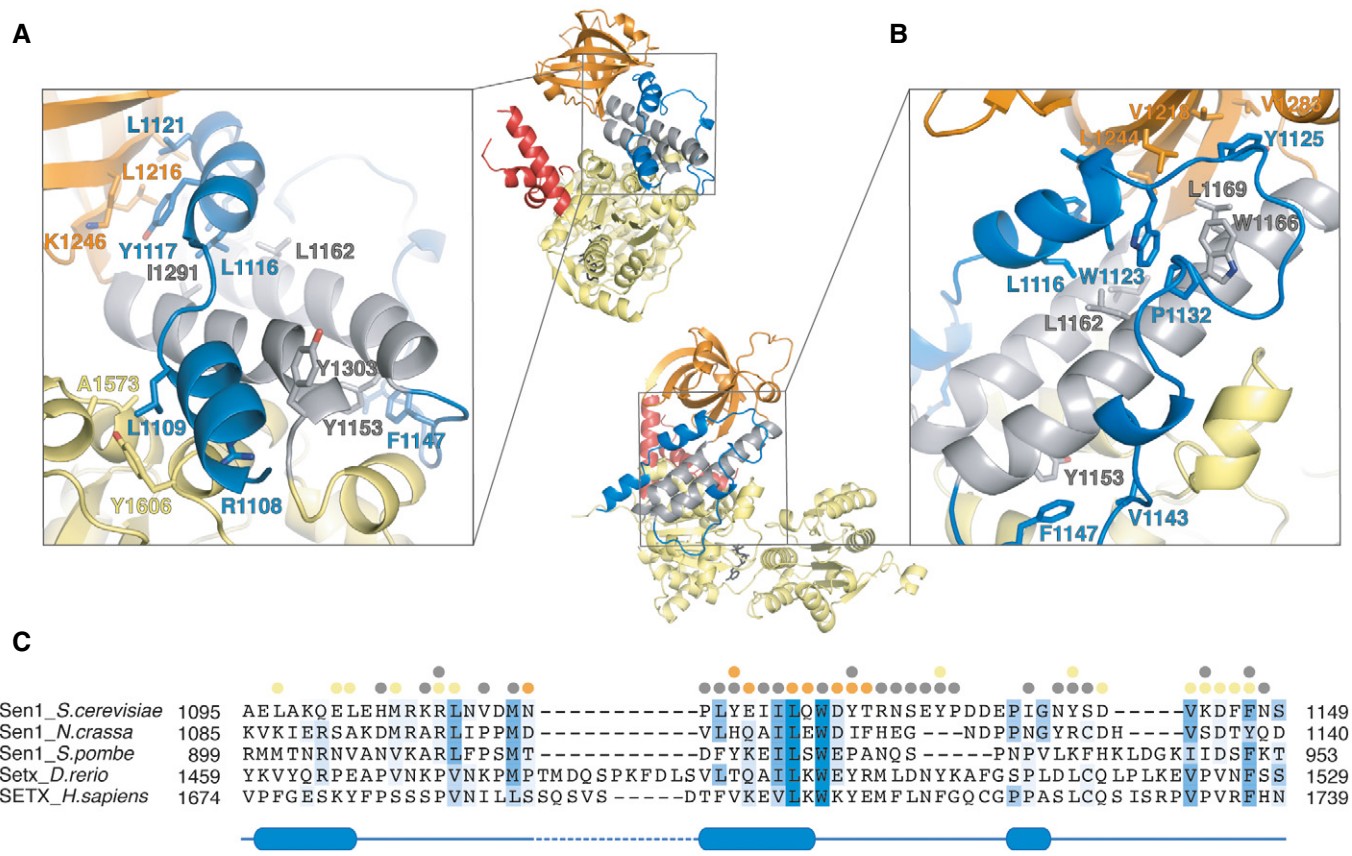

**Figure 3.  The evolutionarily conserved interactions of the N-terminal "brace" of Sen1.**

A, B  Zoom-in views (with the corresponding overall views) showing the extensive hydrophobic interactions the "brace" makes with RecA1, the "stalk" helices, and the "barrel". Selected residues are shown in stick representation. Panels (A and B) show the molecule in a side-view and back-view orientations (90° and 180° clockwise rotation around a vertical axis with respect to the front-view in Fig 2).

C  Structure-based sequence alignment of the "brace" showing the amino acid conservation (highlighted in blue) and the interactions with RecA1 (yellow circles), the "stalk" helices (gray circles), and the "barrel" (orange circles).

Sen1$_{Hel}$, however, the "barrel" is already displaced from the equivalent RNA-binding surface and adopts the position (albeit not the orientation) observed in the RNA-bound state of Upf1$_{Hel}$.

## The RNA binding properties of Sen1

Generally, the RNA binding interactions of Sen1$_{Hel}$ are expected to be similar to those observed in the structure of Upf1$_{Hel}$-U$_6$-ADP:

AlF$_4^-$ (Chakrabarti *et al*, 2011). Sen1$_{Hel}$ shares the conserved RecA2 residues that interact with ribonucleotides 1 and 2, at the 5′ end of the RNA (Tyr1752$_{Sen1}$ and Arg1813$_{Sen1,}$ corresponding to yeast Tyr732$_{Upf1}$ and Arg794$_{Upf1}$) (Fig EV2C). It also shares residues in the RecA1-RecA2 linker that approaches ribonucleotides 3 and 4, in the central portion of the RNA (Pro1622$_{Sen1}$ and Thr1623$_{Sen1}$, corresponding to Pro604$_{Upf1}$ and Val605$_{Upf1}$). Finally, it shares residues in the RecA1 domain and in the "stalk" that interact with

**Figure 4.  Analysis of RNA binding features of Sen1.**

A  Comparison of the structures of yeast Sen1$_{Hel}$-ADP, human UPF1$_{Hel}$-AMPPNPP (PDB: 2GJK, Cheng *et al*, 2007), UPF1$_{Hel}$-ADP:AlF$_4^-$-RNA (PDB: 2XZO, Chakrabarti *et al*, 2011), and yeast Upf1$_{Hel-CH}$-ADP:AlF$_4^-$-RNA (PDB: 2XZL, Chakrabarti *et al*, 2011). Colors are the same as in Fig 2. The nucleotides and RNA are shown in black. On the bottom, schematic representation of the subdomain organization of Sen1 and Upf1 illustrating the different location of the "barrel" (in orange) and its repositioning in Upf1 upon RNA binding. Note that the CH domain of Upf1 pushes the "barrel" and changes its orientation extending the RNA-interaction region. The molecules in a side-view orientation are shown in Fig EV2B.

B  RNase protection assays with yeast Sen1$_{Hel}$, Upf1$_{Hel}$, and Upf1$_{CH-Hel}$ in the presence of different nucleotides (ADP:BeF$_3^-$ and ADP:AlF$_4^-$, which mimic the ground state and the transition state of the nucleotide in the ATPase cycle). A Coomassie-stained SDS–PAGE with the different proteins used is shown on the left (M: molecular weight marker). RNA fragments were obtained by digesting $^{32}$P body-labeled (CU)$_{28}$C 57-mer RNA in the presence of the indicated proteins with RNase A and RNase T1. The left lane was loaded with 10-mer and 15-mer radioactively labeled transcripts as size markers. The asterisks (*) identify minor fragments likely due to the contiguous binding of more than one protein to the same RNA.

Source data are available online for this figure.

ribonucleotides 5 and 6, at the 3′ end of the RNA (Thr1289_{Sen1}, Arg1293_{Sen1,} and Asn1413_{Sen1} corresponding to Thr356_{Upf1}, Arg360_{Upf1,} and Asn462_{Upf1}) (Fig EV2B). In support of this structural

analysis, a Sen1_{Hel} T1289A, R1293A double mutant was impaired in RNA binding, ATP hydrolysis, and transcription termination *in vitro* (Fig EV2D–F).

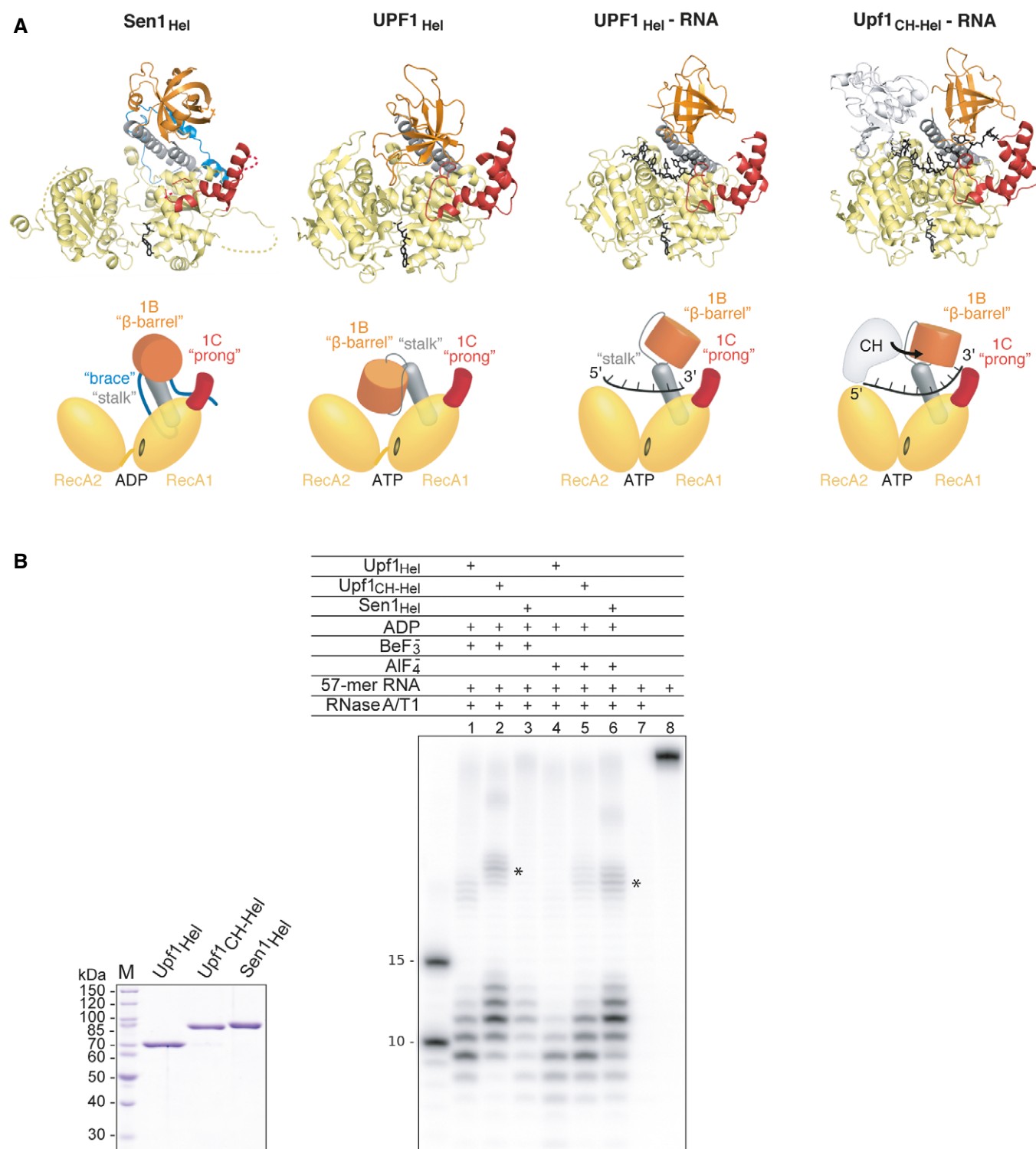

**Figure 4.**

A major difference between Sen1 and Upf1 is that the latter contains an additional CH domain, which regulates RNA binding in an allosteric manner. In $Upf1_{CH-Hel}$, binding of the CH domain onto RecA2 effectively pushes the "barrel" toward the "prong", creating a binding site for two additional ribonucleotides (7 and 8) at the 3′ end of the RNA (Chakrabarti *et al*, 2011) (Fig 4A). No CH domain is present in the sequence of Sen1. We assessed the RNA binding properties of $Sen1_{Hel}$ in RNase protection assays (Fig 4B). As previously reported, $Upf1_{CH-Hel}$ protected longer RNA fragments (~10–11 ribonucleotides) than $Upf1_{Hel}$ (~9 ribonucleotides) (Chamieh *et al*, 2008), consistent with the structural data (Chakrabarti *et al*, 2011). In similar RNase protection assays, $Sen1_{Hel}$ protected 11-ribonucleotide long fragments (Fig 4B), more similar to $Upf1_{CH-Hel}$ than to $Upf1_{Hel}$. This finding raises the question of which intrinsic feature of $Sen1_{Hel}$ mimics the effect of the separate CH domain of $Upf1_{CH-Hel}$ on the RNA footprint. The structural analysis pointed to the "brace", which appears to pull the "barrel" of $Sen1_{Hel}$ toward the "prong" and pre-position it for RNA binding. If our hypothesis is correct, the extended footprint we observed should be directly dependent on the "prong".

In order to test this prediction, we engineered two mutants in $Sen1_{Hel}$ in which we truncated only the upper part of the "prong" that is missing in the present structure ($Sen1_{Hel}\Delta1,471–1,538$ or $Sen1_{Hel}\Delta UP$ for *upper* "prong" deletion) or the entire solvent-exposed portion of the "prong" ($Sen1_{Hel}\Delta1,461–1,554$ or $Sen1_{Hel}\Delta LP$ for *lower* "prong" deletion) (Fig 5A). In RNase protection assays, $Sen1_{Hel}\Delta UP$ had an RNA footprint similar to that of the wild-type protein, but $Sen1_{Hel}\Delta LP$ resulted in a smaller footprint, with the protection of fragments of ~9 ribonucleotides (Fig 5B). This pattern is consistent with our model that the "barrel" and the "prong" of $Sen1_{Hel}$ come together to interact with additional nucleotides at the 3′ end.

## The 5′-3′ RNA-unwinding features of Sen1

As mentioned above, both the SF1 and the SF2 helicases bind RNA with the same directionality across the two RecA domains. In the case of SF2 Ski2-like helicases (which unwind RNA duplexes processively in the 3′-5′ direction), the RNA-unwinding element has been identified as a β-hairpin that protrudes from RecA2, where the 5′ end of a bound RNA resides, and separates the strands of an incoming duplex as it enters the helicase core (Büttner *et al*, 2007; Ozgur *et al*, 2015). For SF1B Upf1-like RNA helicases, which unwind RNAs processively in the opposite

direction (5′-3′), we reasoned that the unwinding element might reside on the opposite side of the helicase to act on an incoming duplex (i.e., near the RecA1 domain). We analyzed the structure of $Sen1_{Hel}$ and compared it to those of $Upf1_{Hel}$, $Upf1_{CH-Hel}$, and $IGBMH2_{Hel}$ to identify a possible structural element on the RecA1 side of the molecule, where the 3′ end of a bound RNA resides. We noticed conserved structural features in the lower part of the "prong" (including $Arg537_{Upf1}$ and $Lys331_{IGHMBP2}$, corresponding to $Arg1552_{Sen1}$) (Fig EV2B). In the $Upf1_{CH-Hel}$ structure, this positively charged residue approaches the very 3′ end of the RNA (Chakrabarti *et al*, 2011).

In order to test a possible role of the "prong" in duplex unwinding, we analyzed the activities of the "prong" mutants described above. Both the $Sen1_{Hel}\Delta UP$ and the $Sen1_{Hel}\Delta LP$ mutants not only retained the footprint in RNase protection assays (Fig 5B), but also retained RNA-dependent ATPase activity (Fig 5C). Deletion of the disordered part of the "prong" in $Sen1_{Hel}\Delta UP$ did not decrease the capacity to dissociate an RNA:DNA duplex, but rather enhanced it. In contrast, the full deletion in $Sen1_{Hel}\Delta LP$ abolished the unwinding activity (Fig 5D). This was not due to a decrease in the affinity for the RNA, since we observed similar RNA binding by $Sen1_{Hel}\Delta LP$ compared to $Sen1_{Hel}$ (Fig EV3). We then assessed the behavior of the same mutants in *in vitro* transcription termination assays. The $Sen1_{Hel}\Delta UP$ mutant exhibited a moderate decrease in termination efficiency. Importantly, the $Sen1_{Hel}\Delta LP$ mutant that was inactive for duplex unwinding was not capable to elicit termination (Fig 5E). Consistent with the major role of the "prong" in termination *in vitro*, in the context of the full-length protein, the LP deletion leads to lethality and provoked major transcription termination defects *in vivo* (Fig EV4). These results indicate that the "prong" is a critical determinant of the 5′ to 3′ unwinding and of the transcription termination activity of Sen1. In the context of termination, the "prong" might have additional functions besides unwinding. When analyzing the structure in detail, we noticed the presence of a hydrophobic residue exposed on the surface of the "prong" and involved in crystal contacts (Leu1549). Reasoning that lattice contacts often occur at protein–protein interaction interfaces, we tested the effect of replacing this residue. We found that the L1549D mutation provoked a less than twofold decrease in $Sen1_{Hel}$ unwinding activity and only a mild decrease in the affinity for the RNA (~twofold), but substantially decreased transcription termination (Figs 5 and EV3). Importantly, a $Sen1_{Hel}\Delta UP$ L1549D double mutant exhibited levels of unwinding activity similar to the wt protein but was strongly affected in transcription termination (Fig 5). These

**Figure 5. A critical role for the "prong" in duplex unwinding and transcription termination.**

A    On top is a schematic presentation of the $Sen1_{Hel}$ variants analyzed in the experiments below. At the bottom is a zoom-in view of the "prong". The dotted lines indicate the approximate positions of the "prong" deletion mutants ΔUP for the removal of the *upper part* and ΔLP for the removal of also *lower part*. Selected residues are shown in stick representation.

B    RNase protection assays with Sen1 proteins as in Fig 4B in the presence of $ADP:AlF_4{}^-$.

C    Analysis of the ATPase activity of the different $Sen1_{Hel}$ variants. Values correspond to the average and SD of three independent experiments.

D    Analysis of the effect of the different "prong" mutations on $Sen1_{Hel}$ unwinding activity. Reaction conditions are the same as in Fig 1B. The graph shows the fraction of duplex unwound as a function of time. Data were fitted with Kaleidagraph to the Michaelis–Menten equation. The values reflect the average and standard deviations (SD) from three independent measurements.

E    IVTT assays in the absence and in the presence of the different $Sen1_{Hel}$ versions (80 nM in the reaction). Representative gel of one out of two to three independent experiments (values of RNA released in additional experiments are provided in the corresponding source data file).

Source data are available online for this figure.

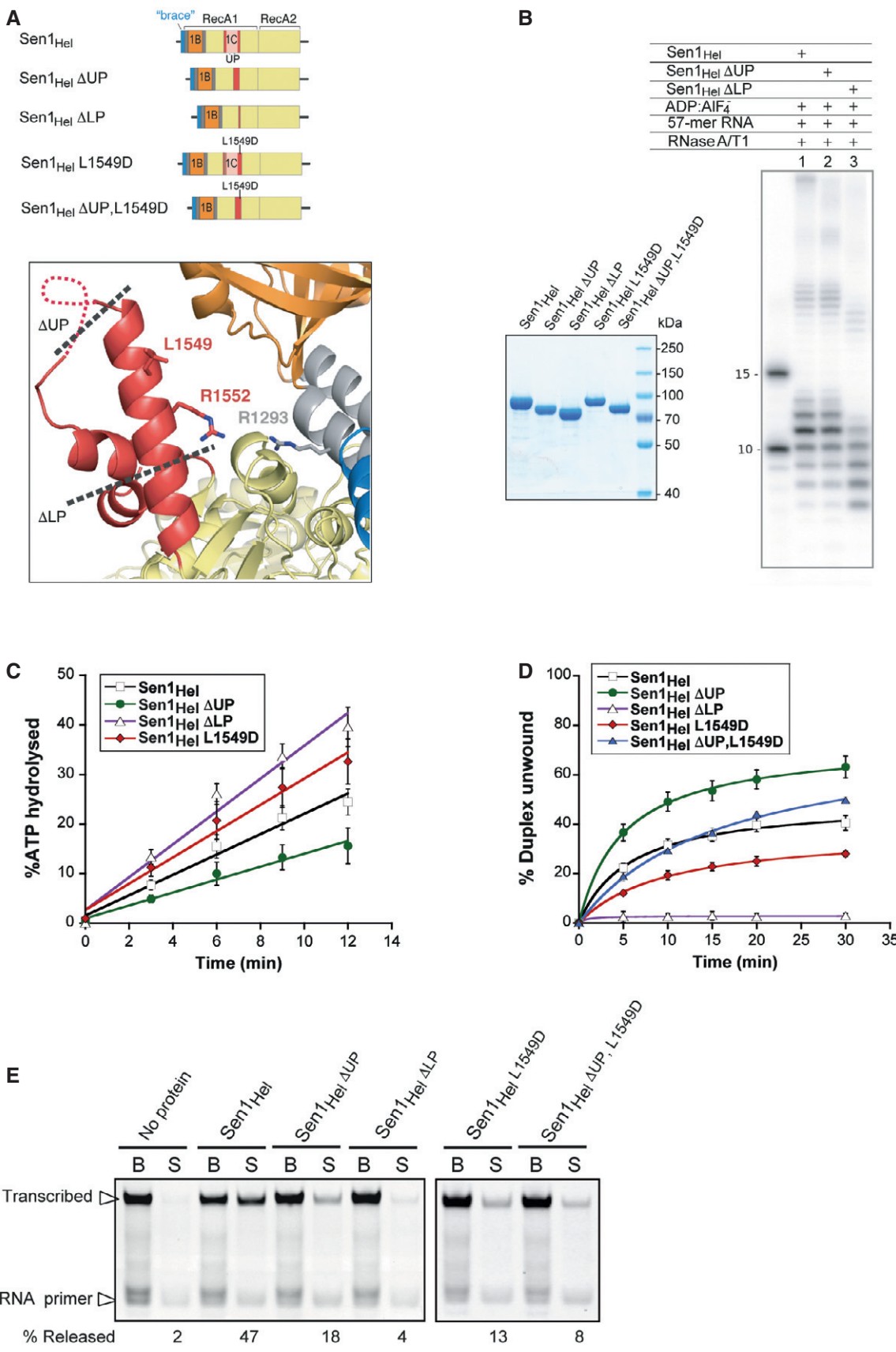

**Figure 5.**

results suggest that the "prong" might mediate protein–protein interactions required for Sen1 transcription termination activity.

## Molecular basis for disease-associated mutations of SETX, the human orthologue of yeast Sen1

Analogously to *S. cerevisiae* Sen1, the human orthologue SETX has been proposed to play crucial roles in transcription termination and in the maintenance of genome integrity (Suraweera *et al*, 2009; Skourti-Stathaki *et al*, 2011; Zhao *et al*, 2016). In line with its biological importance, mutations in SETX have been linked to two neurological disorders: ALS4 and AOA2. Many of the AOA2-causing mutations are missense mutations at the N-terminal domain and at the helicase domain of SETX, which shares about 30% sequence identity with yeast Sen1 (Fig EV5). Importantly, SETX possesses the key residues of the "brace" that are absent in other related helicases (Fig 2). Given the conservation, we took advantage of our Sen1$_{Hel}$ structural data and biochemical tools to get insights into the molecular effects of AOA2-associated mutations in SETX. We mapped 25 missense AOA2 mutations on the Sen1$_{Hel}$ structure (Appendix Fig

S1). Two-thirds of the mutations target residues buried inside the helicase core, and their substitution is expected to disrupt the fold of the protein. A third of the mutations maps near the surface and/or the regions that in other helicases are important for either RNA recognition or ATP hydrolysis, suggesting that these mutations would affect SETX catalytic activity.

In order to test this prediction, we used Sen1$_{Hel}$ as a surrogate for SETX. We introduced a subset of AOA2-associated substitutions at the equivalent positions in the yeast protein (Fig 6A) and analyzed their effect on Sen1$_{Hel}$ activities (Fig 6B–E and Table 2). First, we constructed a disease mutant affecting a residue buried at the interface between the two RecA domains, the Sen1$_{Hel}$ D1616V mutant (D2207V in SETX), which resulted in an insoluble protein (B Leonaite, E Conti, unpublished observations). Second, we analyzed disease mutants expected to affect residues in direct contact with the RNA. The N1413S and T1779P mutants (N2010S and T2373P in SETX, respectively) were impaired in RNA binding, and consequently, duplex unwinding and *in vitro* transcription termination (Fig 6B–D). The effect was similar to the double T1289A, R1293A mutant described before, which affects adjacent residues (Fig EV2C). The

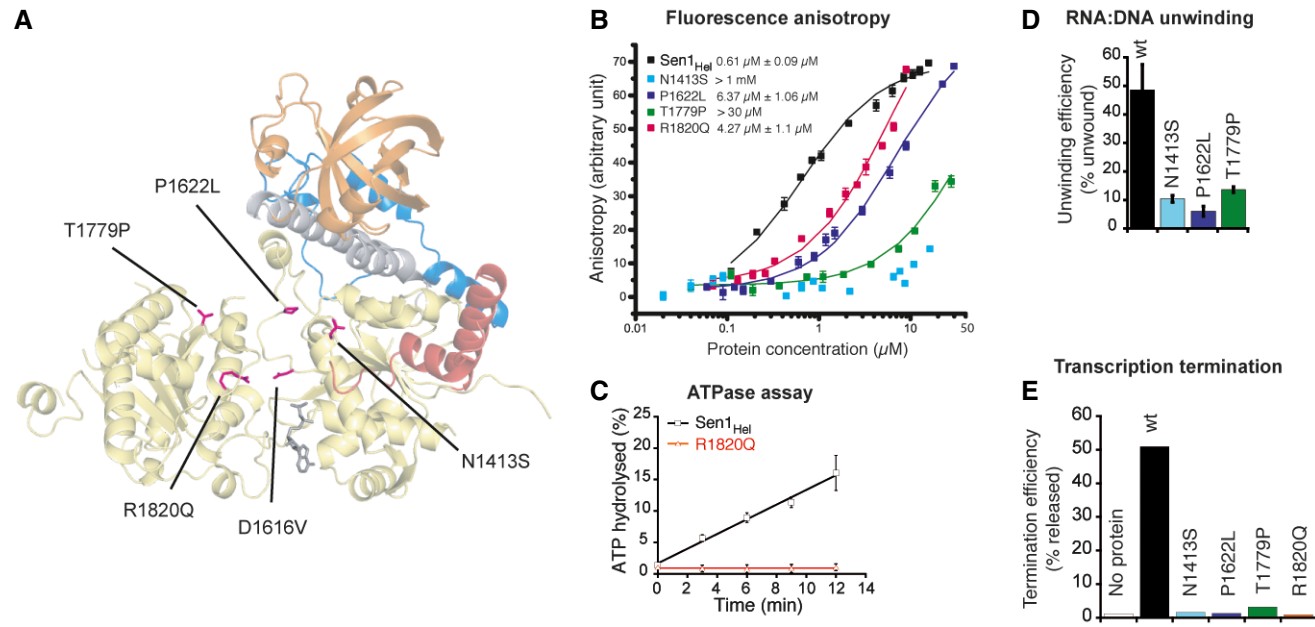

**Figure 6. Functional characterization of Sen1$_{Hel}$ mutants harboring AOA2-associated substitutions.**

A   Mapping of selected AOA2-associated substitutions (shown in magenta) introduced at the equivalent positions in Sen1$_{Hel}$ for their functional analysis. The mutations are reported in Chen *et al* (2014) and in the UCLA Neurogenetics SETX Database.

B   Quantitative measurements of RNA binding affinities of the mutants by fluorescence anisotropy using fluorescently labeled AU-rich RNA as the substrate. The data were fitted to a binding equation describing a single-site binding model to obtain the dissociation constants ($K_D$, indicated on the left of the curves). The best fit was plotted as a solid line. The $K_D$s and their corresponding errors are the mean and standard deviation (SD) of a minimum of three independent experiments.

C   Analysis of the ATPase activity of the Sen1$_{Hel}$ R1820Q mutant predicted to be affected in nucleotide binding. Values correspond to the average and SD of three independent experiments.

D   Assessment of the effect of several AOA2-associated mutations on Sen1$_{Hel}$ unwinding activity. RNA:DNA duplex unwinding reactions using a 75-mer RNA annealed to a 20-mer DNA oligonucleotide (see Appendix Table S1) leaving a 5′-end 55 nt single-strand overhang as the substrate. Reactions contained 30 nM of Sen1$_{Hel}$ variants and 0.5 nM of substrate. The efficiency of unwinding indicated corresponds to the fraction of substrate unwound by the different proteins at 30 min. Values correspond to the average and SD of three independent experiments.

E   Analysis of the impact of AOA2-associated mutations on the efficiency of transcription termination. IVTT assays performed in the presence of 80 nM of the different Sen1 variants. The values of nascent RNA released correspond to one out of two independent experiments. Quantification of both experiments is included in the corresponding source data file.

Source data are available online for this figure.

**Table 2.   Summary of the phenotypes of Sen1$_{Hel}$ mutants.**

| Sen1$_{Hel}$ version | ATP hydrolysis | RNA binding | Unwinding | Termination | Growth[a] |
|---|---|---|---|---|---|
| Wild type | +++ | +++ | +++ | +++ | Normal |
| T1289A, R1293A | +/− | +/− | ND | − | ND |
| N1413S | ND | − | +/− | − | HS |
| Δ1471–1538 (ΔUP) | ++ | +++ | ++++ | ++ | ND |
| Δ1461–1554 (ΔLP) | +++ | +++ | − | − | Lethal |
| L1549D | +++ | ND | ++ | +/− | ND |
| ΔUP, L1549D | ND | ND | +++ | +/− | ND |
| E1591Q | − | ND | ND | − | ND |
| P1622L | ND | + | − | − | Lethal |
| T1779P | ND | +/− | +/− | +/− | HS |
| R1820Q | − | +++ | ND | − | Lethal |

ND, not done; HS, heat sensitive.
[a]Growth of yeast expressing the indicated version of Sen1 according to Chen *et al* (2014), except for ΔLP (see Fig EV4).

P1622L mutant (P2213L in SETX), which substitutes a residue expected to be about 10 Å away from the RNA, resulted in a significant decrease in RNA binding affinity ($K_D$ of ~6.4 μM, as compared to the ~0.6 μM for the wild-type protein). Accordingly, this mutant was also deficient in unwinding and transcription termination (Fig 6D and E). Last, we analyzed a disease mutant predicted to involve a residue in direct contact with the γ-phosphate of ATP and therefore expected to be specifically impaired in ATP hydrolysis, the R1820Q mutant (R2414Q in SETX). As predicted, this mutant did not exhibit any detectable ATPase activity. Because transcription termination strictly depends on ATP hydrolysis, the R1820Q mutant was dramatically affected in this activity (Fig 6B–E). Taken together, our structure–function analyses support the idea that Sen1 is a good model to study the properties of SETX and the molecular basis of the diseases provoked by its mutation.

## Conclusions

RNA helicases of the Upf1-like family take part in a variety of biological functions. Common to all Upf1-like helicases is the ability to unwind nucleic acids in the 5′-3′ direction. We propose that this property is based on a common molecular mechanism of unwinding that depends on the presence of similar structural elements. In all members of the Upf1-like family studied to date, subdomain 1B (the "stalk" and the "barrel") and subdomain 1C (that we refer to as the "prong") extend on top of the RecA1 domain, where the nucleic acid is expected to enter the helicase channel. The conserved part of the "prong" has a similar conformation in all structures of SF1B RNA helicases determined to date and thus appears to be a rather rigid element. In contrast, the "barrel" can generally adopt different conformations and respond to RNA binding as well as to direct or indirect regulatory elements. Although the details are still unclear and the regulation is likely to differ in different members of the SF1B family, we envisage a general mechanism whereby closure of the two subdomains around the incoming RNA allows the "prong" to insert into the duplex, melting it. This strand-separation mechanism is significantly different from that proposed for DNA helicases of the SF1A family (Velankar *et al*, 1999). However, it is reminiscent of the 3′-5′ unwinding mechanism proposed for RNA helicases of the SF2 Ski2-like family, where a structural element on the RecA2 domain (the β-hairpin) inserts into the duplex and melts the incoming base pairs (Büttner *et al*, 2007). Melting elements on opposite sides of the helicase core together with the movements of the two RecA domains in response to ATP hydrolysis could thus underpin the opposite unwinding polarities of Upf1-like and of Ski2-like RNA helicases.

Notwithstanding the similarities in unwinding properties, different RNA helicases of the SF1B family have different biological functions. Sen1 has a specific function in termination of non-coding transcription in yeast cells (Steinmetz *et al*, 2006) and the endogenous full-length protein can indeed recapitulate transcription termination in reconstituted *in vitro* assays (Porrua & Libri, 2013). The most characteristic feature of Sen1 is the presence of a large N-terminal domain, which is important for Sen1 function *in vivo* (Ursic *et al*, 2004). However, we have found that the helicase core of Sen1 retains all the properties that are necessary for transcription termination *in vitro* (Fig 1). Although in an *in vivo* situation the additional domains are likely involved in the recruitment of Sen1 near the nascent RNA and/or regulation, the specificity for the termination reaction *in vitro* is embedded in the helicase core alone. One possibility is that specific features on the outer surfaces of the Sen1 helicase core might contact Pol II. Another, not necessarily exclusive, possibility is that specificity determinants are in the unwinding elements of this helicase.

The region of Sen1 where unwinding is expected to occur and where Pol II is expected to come in close proximity has indeed unique features, with an additional structural element (that we dubbed the "brace") that encircles the RecA1 domain, the "barrel", and the "stalk" into a rather rigid unit. Residues in the "brace" are important *in vivo* (Chen *et al*, 2014) and are highly conserved in Sen1 orthologues, including the human SETX, further supporting the importance of this region. From the structural and biochemical analysis, the "brace" appears to push the "barrel" in a position competent for RNA binding, similar to the effect exerted by the presence of the CH domain in Upf1. However, the "brace" does not lock

Sen1 on the RNA as the CH domain does. The helicase core of Sen1 is indeed active in unwinding, similar to Upf1 when the CH domain has been displaced. We propose that the particular conformation that the "brace" imposes on the accessory domains as well as the distinctive characteristics of the "prong", which we have shown to be essential for termination both *in vitro* and *in vivo*, are key determinants of the specific function of Sen1 in transcription termination. The current model is that Sen1 bind the nascent RNA and translocate along it until it encounters Pol II. Given that the integrity of the "prong" is essential for dissociation of the elongation complex, it is tempting to speculate that the final step of termination involves the insertion of the "prong" into the Pol II RNA exit channel, which would lead to profound conformational changes and destabilization of the elongation complex. This process might be facilitated by the flexible nature of the upper part of the "prong" of Sen1 and/or by protein–protein interactions between the "prong" and specific surfaces of Pol II. Finally, all disease mutations of human SETX we mapped on yeast Sen1 resulted in a transcription termination defect *in vitro*, supporting the idea that the development of AOA2 might be associated with transcription termination defects.

# Materials and Methods

### Protein expression and purification

For Sen1$_{Hel}$ expression, we used a fusion protein with a cleavable C-terminal His tag coupled to *Vibrio cholerae* MARTX toxin cysteine protease domain (His$_8$-CPD) (Shen *et al*, 2009). His$_8$-CPD tagged Sen1$_{Hel}$ (1,095–1,904) and its mutant derivatives were purified from *Escherichia coli* BL21 (DE3) STAR pRARE (Stratagene) cells grown in TB medium. Overexpression was induced by adding IPTG (0.5 mM final concentration) at 18°C overnight. Cells were lysed in buffer containing 20 mM sodium phosphate pH 8.0, 500 mM NaCl, 2 mM MgCl$_2$, 30 mM imidazole, 10% (v/v) glycerol, 1 mM β-mercaptoethanol, benzonase, and protease inhibitors. The proteins were bound to a Ni$^{2+}$-affinity chromatography column (HisTrap FF from GE Healthcare) and eluted by on-column tag cleavage using the 3C protease (Youell *et al*, 2011). CPD tag cleavage with 3C allowed us to overcome an unspecific cleavage product we otherwise obtained when using inositol hexakisphosphate (InsP6). The eluates were then subjected to heparin affinity chromatography on a HiTrap Heparin HP column (GE Healthcare) using buffer A for binding (20 mM Tris–HCl pH 7.5, 200 mM NaCl, 2 mM MgCl$_2$, 1 mM DTT) and buffer B for elution (20 mM Tris–HCl pH 7.5, 1 M NaCl, 2 mM MgCl$_2$, 1 mM DTT). Size-exclusion chromatography (SEC) was performed as a final step of purification using a Superdex 200 column (GE Healthcare) and elution buffer containing 20 mM HEPES pH 7.5, 300 mM NaCl, 2 mM MgCl$_2$, and 1 mM DTT. Proteins were stored at −80°C on SEC buffer containing 50% (v/v) glycerol.

TAP-tagged full-length Sen1 was overexpressed from the *GAL1* promoter in the presence of galactose in *S. cerevisiae* (strain YDL2556) and purified using a previously described protocol (Porrua & Libri, 2015b) with the following modifications: the concentration of NaCl in elution buffers was increased to 500 mM to improve the elution yield, and proteins bound to IgG-beads were treated with 20 μg/ml of RNase A during elution at 4°C overnight.

RNA pol II (12 subunits) was purified from *S. cerevisiae* strain BJ5464 (Kireeva *et al*, 2003) by Ni$^{2+}$-affinity chromatography followed by anion exchange essentially as previously described (Porrua & Libri, 2015b). Recombinant His$_6$-tagged Rpb4/7 heterodimer was purified Ni$^{2+}$-affinity chromatography and gel filtration as previously described (Porrua & Libri, 2015b).

### Crystallization and structure determination

Sen1$_{Hel}$ was concentrated to 3 mg/ml and mixed with a 10-fold molar excess of ADP. Crystals were grown at 4°C by hanging-drop vapor diffusion from drops formed by equal volumes of protein and of crystallization solution (6% (w/v) PEG 8000, 8% (v/v) ethylene glycol, 0.1 M HEPES pH 7.5). Prior to flash freezing in liquid nitrogen, the crystals were briefly soaked in mother liquor containing 28% (v/v) ethylene glycol. The best diffracting crystals were obtained by removing a disordered loop (1,471–1,538) with a (Gly-Ser)$_2$ linker.

A single-wavelength anomalous diffraction experiment from intrinsic sulfur atoms (S-SAD) was performed at the macromolecular crystallography super-bending magnet beamline X06DA (PXIII) at the Swiss Light Source (Villigen, Switzerland). On a single crystal, 4 × 360° data sets were collected at 100 K at a wavelength of 2.075 Å with 0.1° oscillation 0.1 s exposure in four different orientations of a multi-axis goniometer (Waltersperger *et al*, 2015), as previously described (Weinert *et al*, 2015). The sample-to-detector distance was set to 120 mm. The data were processed using XDS and scaled and merged with XSCALE (Kabsch, 2010). The high-resolution data cutoff was based on the statistical indicators CC1/2 and CC* (Karplus & Diederichs, 2012). Substructure determination and phasing were performed with SHELXC/D/E (Sheldrick, 2010) using the HKL2MAP interface (Pape & Schneider, 2004). The successful SHELXD substructure solution, in a search for 25 sulfur sites, had a CCall and a CCweak of 36.9 and 18.2, respectively. Density modification resulted in a clear separation of hands. Three cycles of chain tracing resulted in the automatic building of 275 amino acids with SHELXE. An initial model was built automatically with BUCCANEER (Cowtan, 2006) and extended manually in the experimental electron density in COOT (Emsley & Cowtan, 2004) and refined against the native data with phenix.refine (Adams *et al*, 2010). The final model includes residues 1,096–1,875, with the exception of missing or disordered loops in subdomain 1C (residues 1,471–1,543), in RecA1 (residues 1,382–1,395), and RecA2 (residues 1,705–1,713 and 1,799–1,801).

### ATP hydrolysis assays

ATPase assays were performed as previously described (Porrua & Libri, 2015b). Briefly, 5 nM of purified Sen1 proteins was assayed at 28°C in 10-μl reactions containing 10 mM Tris–HCl pH 7.5, 75 mM NaCl, 1 mM MgCl$_2$, 1 mM DTT, 25% glycerol, and 50 ng/μl polyU. The reaction started with the addition of a 250 μM ATP solution containing 0.25 μM of 800 Ci/mmol α$^{32}$P-ATP (final concentrations). Aliquots were taken at various times, mixed with one volume of quench buffer (10 mM EDTA, 0.5% SDS), and subjected to thin-layer chromatography on PEI cellulose plates (Merck) in 0.35 M potassium phosphate (pH 7.5). Hydrolysis products were analyzed by phosphorimaging (GE Healthcare).

## Duplex unwinding assays

The RNA:DNA substrates for the unwinding assays were formed by annealing a short 5′-end labeled DNA oligonucleotide to either the 5′ or 3′ end of a longer RNA oligonucleotide (Appendix Table S1). The 44-mer RNA oligonucleotide was purchased from Integrated DNA Technologies, whereas the 75-mer RNA was produced by *in vitro* transcription with the appropriate templates (see Appendix Table S1) using the MEGAshortscript T7 kit (Ambion). Unwinding assays were performed in unwinding buffer (10 mM Tris–HCl pH 7.5, 50 mM NaCl, 7.5 μM ZnCl₂, 0.5 mM DTT, 10% glycerol, 0.1 mg/ml BSA) in 20-μl reactions at 28°C. Sen1 proteins were preincubated with the corresponding duplex substrate, and the reaction was initiated by adding a mixture containing ATP and MgCl₂ (2 mM in the reaction) and an excess of unlabeled DNA oligonucleotide (0.1 μM) to trap the unwound RNA. Aliquots were taken at the indicated time-points and mixed with 1 volume of stop/loading buffer containing 50 mM EDTA, 1% SDS, and 20% glycerol. Samples were subjected to electrophoresis on a 15% native PAGE, and gels were directly scanned using a Typhoon scanner (GE Healthcare).

## *In vitro* transcription termination assays

Termination assays were performed basically as previously described (Porrua & Libri, 2015b). Briefly, ternary ECs were assembled in a promoter-independent manner by first annealing a fluorescently labeled RNA (oligo DL2492, see Appendix Table S1) with the template DNA (oligo DL3352, see Appendix Table S1) and subsequently incubating the RNA:DNA hybrid with purified RNA pol II. Next, the non-template strand (oligo DL3353, see Appendix Table S1) and recombinant Rpb4/7 heterodimer were sequentially added to the mixture. The ternary ECs were then immobilized on streptavidin beads (Dynabeads MyOne Streptavidin T1 from Invitrogen) and washed with transcription buffer (TB) containing 20 mM Tris–HCl pH 7.5, 100 mM NaCl, 8 mM MgCl₂, 10 μM ZnCl₂, 10% glycerol, and 2 mM DTT; then with TB/0.1% Triton, TB/0.5 M NaCl, and finally TB. The termination reactions were performed at 28°C in TB in a final volume of 20 μl in the absence or in the presence of 20–80 nM of Sen1 proteins. Transcription was initiated after addition of a mixture of ATP, UTP, and CTP (1 mM each as the final concentration in the reaction) to allow transcription through the G-less cassette up to the first G of a G-stretch in the non-template strand. The reactions were allowed for 15 min and then stopped by the addition of 1 μl of 0.5 M EDTA. After separation of beads and supernatant fractions, beads fractions were resuspended in 8 μl of loading buffer (1× Tris-borate-EDTA, 8 M urea) and boiled for 5 min at 95°C, while RNAs in the supernatant fractions were ethanol precipitated and resuspended in 8 μl of loading buffer. Transcripts were subjected to 10% (w/v) denaturing PAGE (8 M urea), and gels were scanned with a Typhoon scanner.

## RNase protection assays

Proteins (10 pmol each) were mixed with 5 pmol ³²P body-labeled RNA to a final 20 μl reaction volume in 50 mM HEPES pH 6.5, 150 mM NaCl, 1 mM magnesium diacetate, 10% (v/v) glycerol, 0.1% (w/v) NP-40, and 1 mM DTT. After incubation for 1 h at 4°C, the reaction mixtures were digested with 1 μg RNase A/T1 mix and 2.5 U RNase T1 (Fermentas) for 20 min at 20°C. Protected RNA fragments were then extracted twice with phenol:chloroform:isoamyl alcohol (25:24:1 (v/v), Invitrogen), precipitated with ethanol, separated on a denaturing 22% (w/v) polyacrylamide gel, and visualized by phosphorimaging (Fuji).

## Fluorescence anisotropy

Fluorescence anisotropy measurements were performed with a 5′-end fluorescein-labeled 15-mer RNA (oligo ARE, see Appendix Table S1) at 20°C in 50-μl reactions on Infinite M1000 Pro (Tecan). The RNA was dissolved to a concentration of 10 nM and incubated with Sen1$_{Hel}$ variants at different concentrations in a buffer containing 20 mM Hepes pH 7.5, 300 mM NaCl, 2 mM MgCl₂, and 1 mM DTT. The excitation and emission wavelengths were 485 nm and 535 nm, respectively. Each titration point was measured three times using ten reads. The data were analyzed by nonlinear regression fitting using the BIOEQS software (Royer, 1993).

## Accession numbers

The coordinates and structure factors of Sen1$_{Hel}$ have been deposited in the Protein Data Bank with the accession code 5MZN.

Additional methods (electrophoretic mobility shift assays and *in vivo* RNA expression analyses) are included in the Appendix Supplementary Methods.

**Expanded View** for this article is available online.

## Acknowledgements

We thank the Max Planck Institute of Biochemistry (MPIB) Crystallization and Core Facilities; Claire Basquin for FA measurements; Vincent Olieric at PXIII at SLS for assistance with data collection and members of our labs for useful discussions. We also thank David Brow (University of Wisconsin) for critical reading of the manuscript and suggestions and Christian Biertumpfel (MPI Biochemistry) for the His₈-CPD expression vector. This study was supported by the Graduate School of Quantitative Biosciences Munich to B.L., by the Max Planck Gesellschaft, the European Commission (ERC Advanced Investigator Grant 294371) and the Deutsche Forschungsgemeinschaft (DFG SFB646, SFB1035, GRK1721, and CIPSM) to E.C.; by the CNRS, the Agence National pour la Recherche (ANR-08-Blan-0038-01 and ANR-12-BSV8-0014-01 to D.L. and ANR-16-CE12-0001-01 to O.P.) and the Fondation pour la Recherche Medicale (Programme Equipes 2013 to D.L.). Z.H. was supported by PhD fellowships from the China Scholarship Council and La Ligue Contre le Cancer.

## Author contributions

BL solved the structure with help from JB and purified all proteins; ZH carried out the enzymatic assays and *in vivo* analysis; FB carried out the RNase protection assays; EC, OP, and DL initiated the project; EC, OP, and BL wrote the manuscript.

## Conflict of interest

The authors declare that they have no conflict of interest.

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
