## [Review Process File · The EMBO Journal]

Manuscript EMBO-2016-96174

Sen1 has unique structural features grafted on the architecture of the Upf1-like helicase family

Bronislava Leonait, Zhong Han, Mr. Jerome Basquin, Mr. Fabien Bonneau, Domenico Libri, Odil Porrua and Elena Conti

Corresponding authors: Odil Porrua, Institut Jacques Monod; Elena Conti, Max Planck Institute of Biochemistry

Review timeline:	Submission date:	25 November 2016
	Editorial Decision:	13 January 2017
	Revision received:	28 February 2017
	Editorial Decision:	28 February 2017
	Revision received:	06 March 2017
	Accepted:	09 March 2017

Editor: Anne Nielsen

Transaction Report:

1st Editorial Decision 13 January 2017

Thank you for submitting your manuscript for consideration by the EMBO Journal and my apologies for the extended duration of the review period, brought on by the recent holidays. Your study has now been seen by three referees and their comments are shown below.

As you will see from the reports, all three referees highlight the technical quality of your work and express interest in the findings reported in your manuscript. They consequently support publication here, pending minor revision that largely relates to text changes.

Given the referees' positive recommendations, I would like to invite you to submit a revised version of the manuscript, addressing the comments of all three reviewers. I should add that it is EMBO Journal policy to allow only a single round of revision, and acceptance of your manuscript will therefore depend on the completeness of your responses in this revised version.

When preparing your letter of response to the referees' comments, please bear in mind that this will form part of the Review Process File, and will therefore be available online to the community. For more details on our Transparent Editorial Process, please visit our website: http://emboj.embopress.org/about#Transparent_Process

We generally allow three months as standard revision time. As a matter of policy, competing manuscripts published during this period will not negatively impact on our assessment of the conceptual advance presented by your study. However, we request that you contact the editor as

soon as possible upon publication of any related work, to discuss how to proceed. Should you foresee a problem in meeting this three-month deadline, please let us know in advance and we may be able to grant an extension.

Thank you for the opportunity to consider your work for publication. I look forward to your revision.

REFeree REPORTS

Referee #1:

This is a nice, simple paper that presents a structure for the Sen1 helicase C-terminal region, which is sufficient for helicase and termination activities. The structure shows the expected two RecA domains and some accessory domains. Comparison to the related Upf1 protein shows that the orientation of the accessory domains are different and may be fixed by a "brace". Based on some RNase footprinting biochemical experiments, it is proposed that the "prong" is involved in termination and in melting RNA-DNA hybrids. Finally, some disease-causing point mutations found in the human senataxin protein are made in Sen1 and shown to affect folding or activity. Overall, there are important insights into Sen1 provided by the paper that will be helpful to everybody working on Sen1 and the entire class of RNA helicases.

A few minor points:

1. It's reasonable to speculate about a role for the prong in unwinding and termination. However, I wonder if the loss of activity in the LP deletion could be because of reduced RNA affinity (as expected from the smaller footprint) rather than a direct effect on the enzymatic steps. Although the L1549D point mutant argues against this, it could be worth doing the RNA binding titration (as in Fig 6) for these mutants to rule it out.
2. I'm confused about one part of the model in the Discussion. It's clear why the unwinding module would be on opposite ends of the RNA channel depending on the helicase polarity, but I don't understand the claim that this position could itself be "responsible" for the polarity of the unwinding. Since the movement along RNA is presumably due to the movement of the two RecA modules, how would the prong or CH domain change the direction? If the authors stand by this proposal, maybe an additional diagram would be useful for illustrating how they think this works.

Referee #2:

The manuscript by Leonaite et al. analyses the structure of the Sen1 helicase from *S. cerevisiae*, which is compared with Upf1 and IGHMBP2, other members of the Superfamily 1B. Based on the structure, the authors perform biochemical and mutational studies, to correlate the structure of Sen1 and its functional properties. In addition, disease-causing mutations in senataxin are interpreted with the structural information provided by the yeast ortholog.

Overall, the work is well performed and provides novel information about the structural bases of Sen1 function in transcription termination. The comparison between the three helicases, their structure and functional properties, provides an interesting perspective to correlate common and specific functions of each protein with their structure.

Minor points

- Despite the interest of the manuscript, the main message the authors want to provide (how the structure of Sen1 illuminates on how/why Sen1 can perform a distinct function in transcription termination) is lost throughout the text, and the discussion could be improved so that this message (why/how) is the main take home message.
- Related with this, some parts of the Results section provide too many structural details, and this is

sometimes difficult to follow. Maybe some simplification, without affecting the core of the description of results, could help readers with interest in helicases but who are not expert structural biologists.

Referee #3:

Conti and co-workers describe the crystal structure of a fragment (~ 44%) of the yeast RNA helicase Sen1. The crystallized fragment contains the helicase core, the beta-barrel domain, and the small stalk and prong domains, all of which are typical for SF1 RNA helicases. The Sen1 fragment also reveals a distinct "brace" segment, which appears to be conserved among eukaryotic Sen1 orthologs. The authors further show that the crystallized Sen1 fragment catalyzes unwinding and transcription termination *in vitro* and they probe the functional significance of several identified structural domains and residues. Finally, they test the functional importance of residues in human SETX that are mutated in diseases and that can now be mapped directly on the Sen1 structure.

The paper is very well crafted. It is clearly written, structure and biochemical experiments are well integrated, and the presented data are of high quality. One might question the degree of overall conceptual advance of the work, which seems to be limited to the discovery of the brace segment. Most of the SETX mutations could - most likely- be mapped on a homology model of SETX made from Upf1 or IGHMBP2 structures, but having a Sen1 structure is somewhat better. Of course, judgments of conceptual advance are somewhat arbitrary.

With respect to the substance of the paper, this reviewer found nothing of significance warranting augmentation or alteration before publication. The ms. is a solid contribution to our understanding of SF1 RNA helicases in general, and of Sen1 and orthologs in particular.

1st Revision - authors' response

28 February 2017

We are very grateful to the Referees for their positive comments and their constructive points. Below is the point-by-point response with the changes we have implemented to address the specific points raised by Reviewers 1 and 2.

Referee #1:

1. It's reasonable to speculate about a role for the prong in unwinding and termination. However, I wonder if the loss of activity in the LP deletion could be because of reduced RNA affinity (as expected from the smaller footprint) rather than a direct effect on the enzymatic steps. Although the L1549D point mutant argues against this, it could be worth doing the RNA binding titration (as in Fig 6) for these mutants to rule it out.

We have tested the RNA-binding properties of the L1549D mutant and of the entire LP deletion. Removal of the LP region (Sen1_{Hel}D1461-1554, which includes L1549) showed similar RNA-binding properties as compared to the wild-type (**new Figure EV3**). Sen1_{Hel}L1549D showed only a mild decrease (~2-fold) (**new Figure EV3**). These data are consistent with the similar levels of unwinding activity between the mutants and wild type, pointing to a specific effect rather than unspecific loss of RNA binding.

2. I'm confused about one part of the model in the Discussion. It's clear why the unwinding module would be on opposite ends of the RNA channel depending on the helicase polarity, but I don't understand the claim that this position could itself be "responsible" for the polarity of the unwinding. Since the movement along RNA is presumably due to the movement of the two RecA modules, how would the prong or CH domain change the direction? If the authors stand by this proposal, maybe an

additional diagram would be useful for illustrating how they think this works.

We agree with the Reviewer, and changed the statement to: “*Melting elements on opposite sides of the helicase core together with the movements of the two RecA domains in response to ATP hydrolysis could thus underpin the opposite unwinding polarities of Upf1-like and of Ski2-like RNA helicases.*”

Referee #2:

- Despite the interest of the manuscript, the main message the authors want to provide (how the structure of Sen1 illuminates on how/why Sen1 can perform a distinct function in transcription termination) is lost throughout the text, and the discussion could be improved so that this message (why/how) is the main take home message.

We now elaborate more in the final discussion paragraph the key determinants for the specific function of the Sen1 helicase domain in transcription termination, namely the *particular conformation that the “brace” imposes to the accessory domains as well as the distinctive characteristics of the “prong”*. In support of this model, we have included in the revised version of the manuscript *in vivo* data showing that *in the context of the full-length protein the LP deletion lead to lethality and provoked major transcription termination defects in vivo (new Figure EV4)*.

- Related with this, some parts of the Results section provide too many structural details, and this is sometimes difficult to follow. Maybe some simplification, without affecting the core of the description of results, could help readers with interest in helicases but who are not expert structural biologists.

We have streamlined the description of the structure and eliminated many of the structural details from the Results section.

2nd Editorial Decision

28 February 2017

Thank you for submitting a revised version of your manuscript. I have now gone through the point-by-point response you provided to address the referee concerns and I happy to inform you that your study is now in principle ready to be accepted for publication here. However, before we can go on to officially accept your manuscript there are a few editorial issues concerning text and figures that I need you to address in a final revision:

-> Please include a brief conflict of interest statement in the manuscript.

-> Could you please provide a brief title for Table 1?

-> We noticed that you refer to Appendix Figure S1 in the manuscript but that there is no appendix uploaded. Is this supposed to be Figure EV1 instead?

-> Since there is only one file in the Appendix, I would encourage you to label this table as Table EV1 (and to update the callouts in the text accordingly)

-> Please rename the 'Experimental Procedures' in the main manuscript to 'Materials and methods'

-> We noticed that the image resolution for the blot in fig 1C is fairly low and if you have a higher resolution scan of the same gel I would encourage you to use that instead.

-> Please make sure that all figures displaying statistics analysis have information on the number of replicas and the nature of the error bars indicated in the figure legend (currently missing in fig 1B, 5D and 6D). In addition, we noticed that the data in fig 6E is displayed with error bars although the data derives from two independent experiments according to the legend. We generally require that $n \geq 3$ for statistical analysis and I would therefore encourage you to display the two data series here instead.

-> We generally encourage the publication of source data, particularly for electrophoretic gels and blots, with the aim of making primary data more accessible and transparent to the reader. We would need 1 file per figure (which can be a composite of source data from several panels) in jpg, gif or PDF format, uploaded as "Source data files". The gels should be labelled with the appropriate figure/panel number, and should have molecular weight markers; further annotation would clearly be useful but is not essential. These files will be published online with the article as a supplementary "Source Data". Please let me know if you have any questions about this policy.

2nd Revision - authors' response

06 March 2017

Thanks again for considering our manuscript for publication in EMBO J. We have just submitted an amended version of our manuscript EMBOJ-2016-96174R containing the following modifications:

1. Manuscript main text:

- Materials and Methods instead of Experimental procedures.
- Statement of no conflict of interest.
- Modified figure legends to include detailed statistical information on experiments in all panels.
- Brief mention to supplementary methods in the Appendix.

2. Modified table 1 including a title.

3. Modified figures 1, 5 and 6 in which the experiments performed in two replicates display the values of one of the replicates (both values are provided in an excel file as source data).

In addition, we have uploaded the following files:

1. Appendix containing Figure S1, Table S1 and supplementary methods.
2. Source data files for figures 1, 4, 5 and 6.

3rd Editorial Decision

09 March 2017

Thank you for submitting the final revision of your manuscript, I am pleased to inform you that it has now been accepted for publication in The EMBO Journal.

Corresponding Author Name: Elena Conti and Odil Porrua

Journal Submitted to: EMBO J

Manuscript Number: EMBOJ-2016-96174